

# Vampyromorph coleoid predation by an ichthyosaurian from the Early Jurassic Lagerstätte of Bascharage, Luxembourg

Valentin Fischer[1,*], Robert Weis[2,*], Dominique François Delsate[2], Francesco Della Giustina[1], Pierre Wintgens[1], Dirk Fuchs[3] and Ben Thuy[2]

[1] Evolution & Diversity Dynamics Lab, Université de Liège, Liège, Belgium
[2] Section Paléontologie, Musée National d'histoire Naturelle, Luxembourg, Luxembourg
[3] Bayerische Staatssammlung für Paläontologie und Geologie, München, Germany
[*] These authors contributed equally to this work.

## ABSTRACT

Many Early Jurassic marine predators were seemingly adapted to hunt soft and fast prey items such as cephalopods. However, deciphering what these animals ate and, therefore, the intensity of their competition is challenging, as fossilised gut content is biased by multiple factors. In this paper, we report a loligosepiid vampyromorph coleoid in the gut of a specimen of the ichthyosaurian *Stenopterygius triscissus* from the early Toarcian Bascharage Lagerstätte of Southern Luxembourg. This is the first report of octobrachian predation in ichthyosaurians. The coeval pachycormid teleosts *Pachycormus macropterus* and *Saurostomus esocinus* have recently been reported to feed on loligosepiid octobrachians as well. We use this opportunity to compare the functional anatomy of these taxa and re-evaluate the affinities of coleoids preserved as ichthyosaurian gut content.

# INTRODUCTION

Over the last 170 years, a series of Lagerstätten deposits have revealed a formidable diversity of marine reptiles that populated the European epicontinental seas during the Toarcian (late Early Jurassic) (*e.g.*, *Hauff, 1953*; *Godefroit, 1994*; *Röhl et al., 2001*; *Großmann, 2007*; *Maisch, 2008*; *Benson et al., 2010*; *Benson et al., 2011*; *Johnson et al., 2018*; *Stöhr & Werneburg, 2022*). These localities indicate the presence of several coeval predators: neoichthyosaurians, thalattosuchian crocodyliforms, and plesiosaurians. Many of them have a long snout and small, acute teeth, suggesting that they relied on soft prey items such as cephalopods and small teleosts (*Massare, 1987*; *Bardet, 1994*; *Godefroit, 1994*; *Fischer et al., 2022a*). Competition among marine reptiles, as well as with other vertebrates such as teleosts or chondrichthyans, is likely, as evidenced in some Mesozoic formations (*Martin et al., 2017*; *Foffa et al., 2018*; *Cortés & Larsson, 2024*). However, precise data on the prey items found in marine reptile gut content is generally lacking. The Bascharage Lagerstätte of Southern Luxembourg has recently revealed that multiple pachycormid teleosts fed

Corresponding authors
Valentin Fischer, v.fischer@uliege.be
Robert Weis, robert.weis@mnhn.lu

on octobranchian cephalopods (*Weis et al., 2024*), while coleoids as a whole are often regarded as a resource consumed by other marine reptiles, notably ichthyosaurians and thalattosuchians based on gut content and dental anatomy (*Massare, 1987*; *Böttcher, 1989*; *Lomax, 2010*). Consumption of cephalopods by ichthyosaurians has been known since the 19th century (*Buckland, 1829*; *Quenstedt, 1858*), based on masses of arm hooks in their fossilised gut contents. Ichthyosaurians have indeed been regarded as 'teuthophages', probably preferring belemnoid coleoids (*Dick, Schweigert & Maxwell, 2016*). Here, we add to this body of evidence by (i) describing the ecomorphology and the gut content of a specimen of the baracromian ichthyosaurian *Stenopterygius triscissus* from the Bascharage Lagerstätte, (ii) comparing it with those of coeval pachycormid teleosts, and (iii) reviewing the systematic attribution of prey items in fossilised ichthyosaurian gut contents.

## MATERIALS & METHODS

### Geological setting

The 'Schistes bitumineux', also known as 'Couches à Harpoceras falciferum' (Lo1) or the Bascharage Lagerstätte, is a geological formation in Luxembourg that crops out over an area approximately 30 km wide between Rodange and Bascharage in the west and Bettembourg and Dudelange to the east (*Dittrich, 1993*); its thickness reaches 40–45 m (*Lucius, 1945*). This formation is the local expression of a widespread phenomenon of anoxia that affected the European epicontinental seas during the early Toarcian, known as the Toarcian Oceanic Anoxic Event (T-OAE) or the Jenkyns Event. As such, the Schistes bitumineux Formation is coeval with other bituminous formations in western Europe, such as the Posidonienschiefer Formation in southwestern Germany, the Jet Rock Formation in the UK, the Schistes Carton in France, and the Grandcourt Formation in Belgium (*Muscente et al., 2023*).

The bituminous shales cropping out in the south of Luxembourg and the neighbouring Gaume and Lorraine regions in France and Belgium are known for their rich fossil record since the 19th century: *Chapuis & Dewalque (1853)* mentioned fossils of 'fish' and 'squids' in Aubange, on the Belgian-Luxembourgish border. Since the 1930s, when two ichthyosaurian skeletons were unearthed (*Faber & De Muyser, 1947*), several specimens have been discovered; see *Godefroit (1994)* for a detailed analysis of this material, as well as *Vincent et al. (in press)*, *Johnson et al. (2018)*, *Laboury et al. (2022)*, and *Bonnevier Wallstedt et al. (2024)* for recent contributions on plesiosaurians, thalattosuchians, and ichthyosaurians, respectively.

The ichthyosaurian specimen studied herein originates from the Schistes bitumineux Formation, more precisely the Serpentinum ammonite zone. It was found during construction works for the 'Luxguard' factory in the industrial zone next to Dudelange (coordinates: 49°30′06″N; 6°04′55″E) by D. Watrinelle, a private fossil collector, in the late '80s or early '90s and subsequently donated to the Palaeontological collections of Musée National d'Histoire Naturelle, Luxembourg (MNHNL) (Fig. 1). The specimen had been described and illustrated as *Stenopterygius quadriscissus* by *Godefroit* (*1994*: pl. 3, fig. c) although we have a slightly distinct interpretation (see below) and is exhibited in the

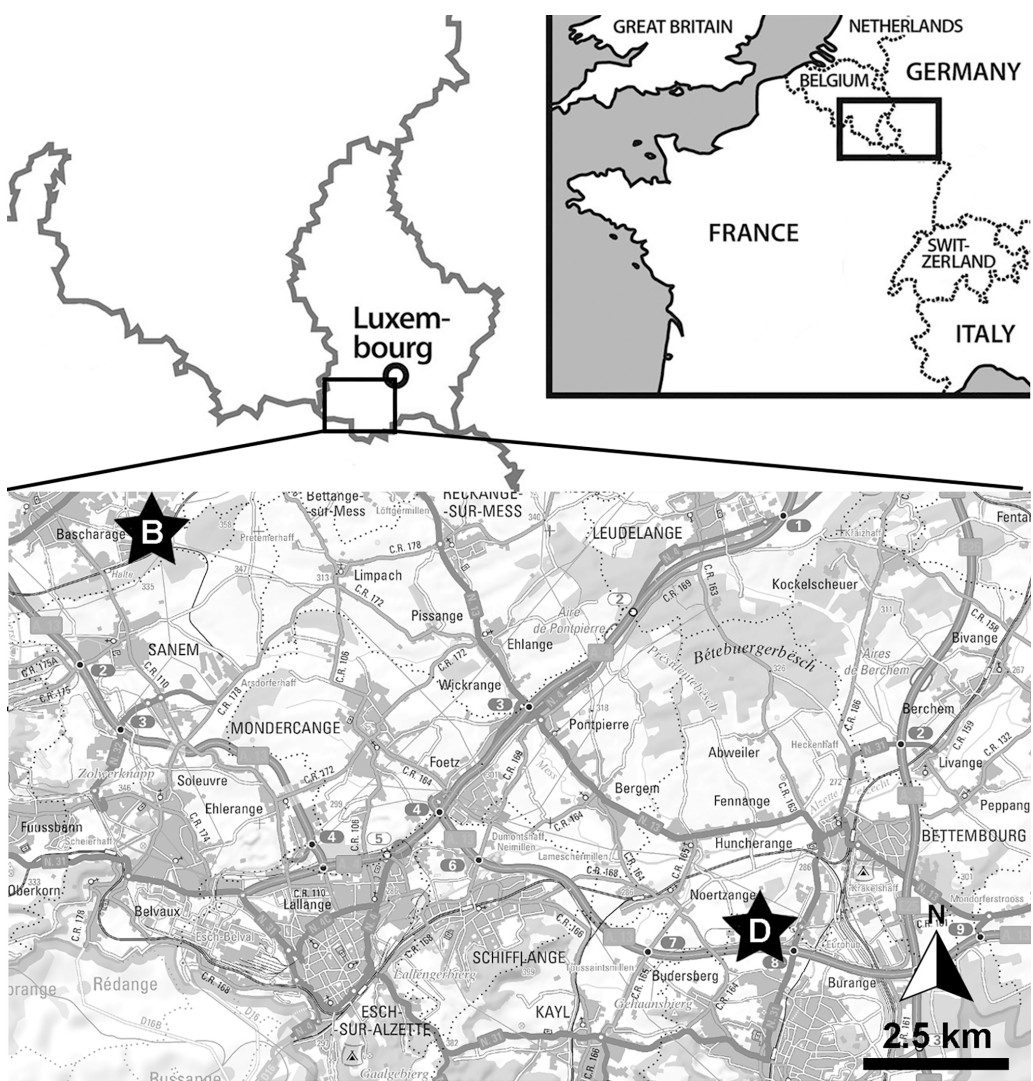

**Figure 1** **Localisation of the Bascharage locality.** Within Western Europe (top-right inset) and within Luxembourg (top-left, bottom). Abbreviations: B, Bascharage Locality; D, Dudelange, another important early Toarcian locality of Luxembourg.

permanent gallery at the MNHNL. A recent restoration by O. Kunze (Stuttgart, Germany) revealed further anatomical details. A 3D model of the full specimen can be accessed freely online: https://sketchfab.com/3d-models/stenopterygius-quadriscissus-tv211-v2024-930e539b4f6e4ef7816d2ad4f4a65227.

## Quantitative taxonomy

*Maxwell (2012)* established a set of linear measurements that can, when used pairwise or in a multivariate analysis, discriminate the species of *Stenopterygius*. This study relied on a homogenous sample of strongly compressed specimens from a single Toarcian formation in southwestern Germany. The preservation in the Schistes bitumineux Formation differs, with more three dimensionality (*e.g.*, *Godefroit, 1994*), but we do not expect this to have

a significant impact on the results. However, the femur, the hindfin, and the ischiopubis dimensions are among the most important for deciphering species in Maxwell's bivariate plots (*Maxwell, 2012*), but are not preserved in MNHNL TV21. In the original study, the linear measurements were log-transformed but were not scaled in any way; as a result, absolute size influences the important axes of the multivariate analysis, and the larger measurements (lower jaw, premaxilla) have more weight than the smaller measurements in driving position on the multivariate plot. Indeed, the first axis of *Maxwell (2012)* concentrates 98.6% of the variance but has to be ignored because of a clear allometric component (*Maxwell, 2012*, p. 110). The influence of larger *vs.* smaller measurements is probably still present in axes 2, 3, 4, *etc.* To solve this and to complement our assessment of the taxonomy of MNHNL TV211 with that morphometric scheme, we downloaded the data from *Maxwell (2012)* and wrote a R script to analyse these data (see Supplementary Information 1 and 3). To remove the effect of absolute size, all measurements were divided by humerus length (which does not seem to be a trait separating species; *Maxwell (2012)*). These new trait ratios were Z-transformed prior to analyses. Two ordination analyses were conducted: a principal coordinate analysis (PCoA), as MNHNL TV211 contains missing information, and a principal component analysis (PCA) restricted to the cranial and forefin traits. Sadly, the improvement in the analytical procedure results in a strong overlap between the species of *Stenopterygius* in principal axes 1 and 2 (no matter the method used); the specimen MNHNL TV211 falls within the convex hulls of both *Stenopterygius quadriscissus* and *Stenopterygius triscissus* in the PCoA and within that of *Stenopterygius quadriscissus* and directly tangent to that of *Stenopterygius triscissus* in the PCA (Fig. S1).

## Functional traits

We gathered a series of craniodental and postcranial traits that summarise the feeding and swimming capabilities of three coeval species, known to have ingested loligosepiid coleoids: the ichthyosaurian *Stenopterygius triscissus* and the pachycormiforms *Pachycormus macropterus* and *Saurostomus esocinus*. We selected the following traits to be compared for each taxon: the absolute total body size (a proxy for energy consumption, among other factors) (*Motani, 2002a*), the mandibular aspect ratio (mandible area divided by the square of mandible length; a proxy for lower jaw stiffness) (*Anderson et al., 2011*; *Friedman, 2012*), the closing mechanical advantage of the jaw, computed at the middle part of the dentigerous section (distance from the quadrate/articular articulation to the paracoronoid process divided by the distance from the quadrate/articular articulation to the mid-dentigerous point in the mandible; a proxy for relative bite strength) (*e.g., Anderson et al., 2011*; *Friedman, 2012*; *MacLaren et al., 2022*), the crown shape (crown height divided by crown basal diameter, for tooth in the middle part of the dentigerous section, a proxy for the piercing capability of the crown) (*e.g., Massare, 1987*; *Fischer et al., 2022a*), the absolute diameter of the opening of the sclerotic ring (a proxy for the size of the dilatated pupil and therefore of low-light vision) (*Nilsson, Warrant & Johnsen, 2014*), and the aspect ratio of the pectoral and caudal fins (fin area divided by the square of the fin proximo-distal length; a proxy for swimming speed). We obtained these data from the literature for pachycormids (*Friedman, 2012*; *Williams, Benton & Ross, 2015*; *Cawley et al., 2019*; *Cooper*

*& Maxwell, 2022*) and from measurements of MNHNL TV211, complemented by the literature (*Lingham-Soliar, 2001*; *Maisch, 2008*) for the ichthyosaurian. The caudal aspect ratio for the ichthyosaurian is taken for the species *Stenopterygius quadriscissus* instead of *Stenopterygius triscissus*, as no specimen of the latter have been reported with an intact caudal fin outline preserved. Articulated apical and postflexural skeletal regions suggest that both species possessed a similarly shaped caudal fin (*Maisch, 2008*; *Maxwell, 2012*). See Supplementary Information 2 and 4) for the data and the R script.

## RESULTS

### Identity and age of the ichthyosaurian

The specimen MNHNL TV211 is a partial ichthyosaurian preserving the skull, neck, torso, scapular girdle, and right anterior forefin, in articulation (total preserved anteroposterior length = 1,047 mm; mandible length = 424 mm) (Fig. 2). The preserved portion suggests that the animal was ≈2,000 mm long *in vivo*. The prefrontal is small and does not participate in the bony narial aperture; the maxilla is anteroposteriorly short (Fig. 2); the teeth are small, straight, and lack conspicuous enamel ornamentation but have apicobasal ridges texturing the root; the coracoid is rounded, with an anterior notch and lacks a posterolateral emargination; the scapula forms a large acromion (at least relative to pre-Middle Jurassic ichthyosaurians); the humerus is constricted at mid-shaft and solely connects distally to two large epipodial hexagonal elements (radius and ulna); the radius and the radiale are notched and the intermedium solely contacts the distal carpal 3. This combination of features is unique to the genus *Stenopterygius* (*Godefroit, 1994*; *Maisch, 2008*; *Caine & Benton, 2011*; *Maxwell, 2012*; *Maxwell & Cortés, 2020*; *Fischer et al., 2022b*).

The rostrum is ≥0.63 the length of the mandible (its anteriormost margin is broken off); the dorsoventral height of the maximal at the level of mid-naris is 0.38 times the dorsoventral height of the rostrum in the same zone; the maxilla forms the anterior part of the ventral border of the bony narial aperture; the trunk appears slender (Fig. 2). This suite of features suggests that the specimen belongs to the species *Stenopterygius triscissus* (*Godefroit, 1994*; *Maisch, 2008*; *Maxwell, 2012*), the most abundant ichthyosaurian species from the Toarcian of Luxembourg (*Godefroit, 1994*). *Godefroit (1994)* reported that the maxilla forms the ventral border of the naris in *Stenopterygius longifrons* (=*Stenopterygius triscissus* (*Maisch, 2008*)), yet a specimen from Strawberry Bank, UK, has the maxilla excluded from the bony narial aperture by a combination of the subnarial process of the premaxilla and the anteroventral process of the lacrimal (*Caine & Benton, 2011*). This feature therefore appears interspecifically variable and should not be used in isolation to identify the species of *Stenopterygius*.

Detailed ontogenetic stages have been described and categorised for *Stenopterygius quadriscissus* (*Miedema & Maxwell, 2019*; *Miedema & Maxwell, 2022*) and can reasonably be applied to other species of the genus. The specimen MNHNL TV211 is likely not a sexually mature specimen: the species is supposed to grow to a size of 3,500 mm while we estimate MNHNL TV211 to be ≈ 2,000 mm long. The internasal and interfrontal connections do not appear to be fully ossified; combined with a mandible length >400

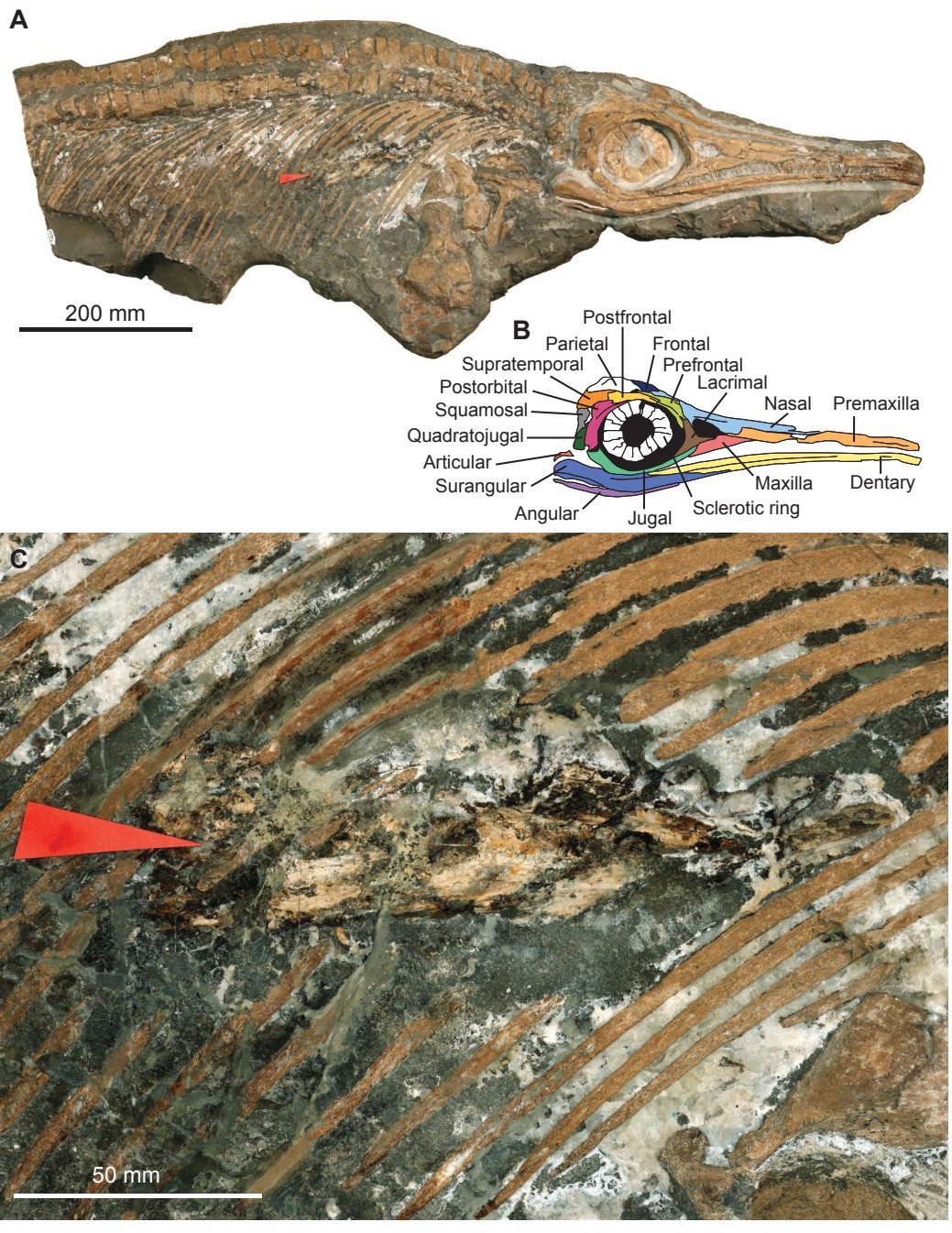

**Figure 2** **Photographs and interpretation of specimen MNHNL TV211.** (A) overview of the specimen. (B) anatomical interpretation of the right lateral portion of the skull. (C) zoom on the fossilised gut content, showing a vampyromorph gladius (red arrow).

mm, this corresponds to the postnatal stage 2 of *Miedema & Maxwell (2022)*. Therefore, the specimen MNHNL TV211 might not have developed all the diagnostic features of the species, notably the long rostrum (70% of total mandible length in adults only, while it is ≥63% in MNHNL TV211) (*Godefroit, 1994*; *Maisch, 2008*). Its juvenile status also

## Comparison of functional traits

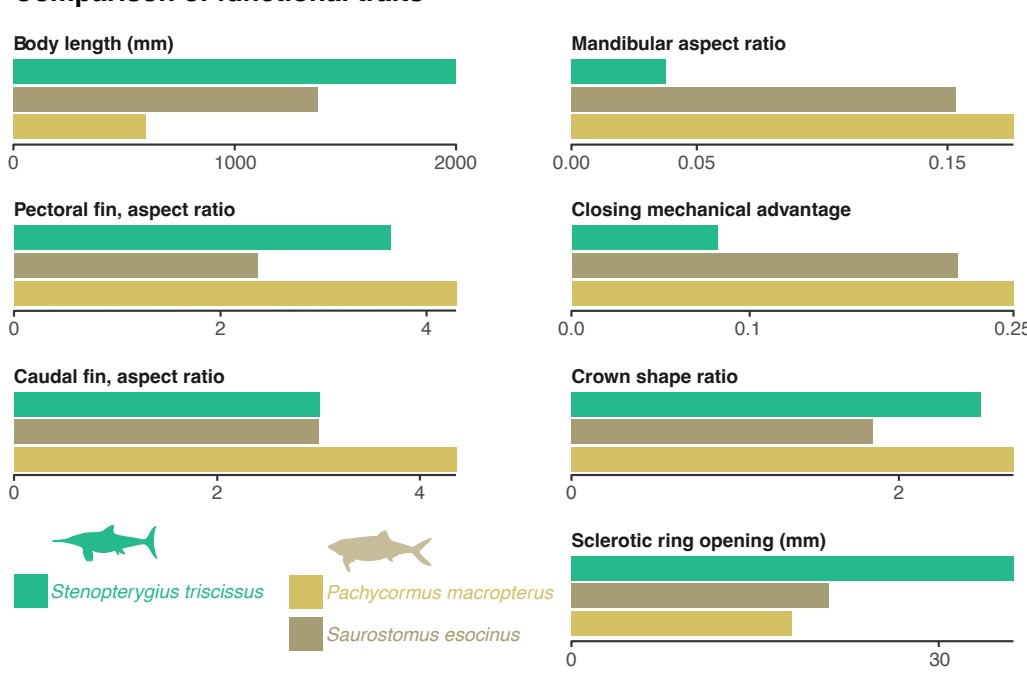

**Figure 3   Comparison of functional traits of the vampyromorphan predators from the Bascharage Laggerstätte.** The absolute total body size is a proxy for energy consumption; the mandibular aspect ratio is a proxy for lower jaw stiffness; the closing mechanical advantage of the jaw is a proxy for relative bite strength; the crown shape is a proxy for the piercing capability of the tooth crown; the absolute diameter of the opening of the sclerotic ring is a proxy for the size of the dilatated pupil and therefore of low-light vision; the aspect ratio of the pectoral and caudal fins are a proxy for swimming speed. See 'Material and Methods' for more details on the computation and justification of these traits, as well as Table 1 and File S2 for metadata and the raw data.

explains the presence of a fully functional dentition. Indeed, *Stenopterygius* is known to lose teeth upon reaching the sexually mature stage; however, the timing and degree of tooth reduction appear to be interspecifically variable (*Godefroit, 1994*; *Maisch, 2008*; *Dick & Maxwell, 2015*). Finally, our multivariate analyses using the data from *Maxwell (2012)* suggest MNHNL TV2011 belong to *Stenopterygius quadriscissus* or *Stenopterygius triscissus* (Fig. S1). We interpret all the evidence presently at hand that MNHNL TV211 is a specimen of *Stenopterygius triscissus*, at postnatal stage 2.

## Functional anatomy of the ichthyosaurian

With an estimated body length of ≈ 2,000 mm, the ichthyosaurian analysed here is about the same size as the common dolphin, *Delphinus delphis* (*Ridgway & Harrison, 1999*). Being an early thunnosaurian ichthyosaurian (*Motani, 1999*), *Stenopterygius* has a thunniform body outline (*Lingham-Soliar & Plodowski, 2007*) and was likely adapted for fast, sustained swimming (*Motani, 2002b*; *Gutarra et al., 2019*). This is also evidenced by the high aspect ratio of the forefin and the caudal fin (Fig. 3).

**Table 1 Functional traits used to compare the vampyromorphan predators from the Bascharage Laggerstätte.** See Supplementary Information 2 for the individual measurements and data sources. Uncited references used: (*Lingham-Soliar, 2001*; *Williams, Benton & Ross, 2015*; *Wretman, Blom & Kear, 2016*).

| Species | Total length | Mandibula aspect | Closing mech advantage | Crown shape | Sclerotic ring opening | Pectoral fin aspect | Caudal fin aspect |
|---|---|---|---|---|---|---|---|
| *Stenopterygius triscissus* | 2,000.00 | 0.04 | 0.08 | 2.44 | 36.10 | 3.65 | 3.01 |
| *Pachycormus macropterus* | 600.29 | 0.18 | 0.25 | 2.70 | 18.00 | 4.28 | 4.34 |
| *Saurostomus esocinus* | 1,374.67 | 0.15 | 0.22 | 1.84 | 21.00 | 2.36 | 3.01 |

The rostrum is long and slender, as in most ichthyosaurians. The paracoronoid process of the surangular (sensu *Bindellini et al., 2021*) is small, as is the postorbital region of the skull. These factors result in a weak bite force; the mechanical advantage computed at the mesialmost tooth of the snout = 0.08 (Fig. 3). Such a value is, for example, one-third to one-half of the value found in most mosasaurids and thalassophonean pliosaurids (*MacLaren et al., 2022*; *Fischer et al., 2016*). The long, slender, and tubular snout of *Stenopterygius triscissus* (mandibular aspect ratio = 0.038; Fig. 3) could be ideal for fast lateral snapping of small prey items, even in the context of reduced head mobility in ichthyosaurians (*VanBuren & Evans, 2016*). The tooth crown in the middle part of the rostrum in MNHNL TV211 has an apicobasal height/basal ratio of 6.1 mm/2.4 mm = 2.5 (Fig. 3) and is therefore slender compared to many other marine amniotes. In *Stenopterygius quadriscissus*, the teeth (when retained) are usually blunter at mandible lengths of ≥400 mm (*Dick, Schweigert & Maxwell, 2016*). The small, narrow crowns of *Stenopterygius triscissus* are straight and lack enamel ornamentation. This suggests a diet of small, soft prey items (*Massare, 1987*; *Fischer et al., 2022a*). The eyes of *Stenopterygius triscissus* are large: the mean diameter of the right orbit of MNHNL TV211 is 91.5 mm, and that of the inner opening of the right sclerotic ring (a direct proxy for the size of the dilatated pupil) is 36.1 mm (Fig. 3), slightly larger than the pupil of the fast swimming swordfish (*Nilsson, Warrant & Johnsen, 2014*). Such a pupil allows a ≈32 m vision range of a 10 cm black target at a depth of 350 m in clear oceanic waters (*Nilsson, Warrant & Johnsen, 2014*).

## Identity of the gut content in MNHNL TV221

The preserved gut content corresponds to the fragmentary remains of a gladius-bearing coleoid, more precisely a loligosepiid octobranchian. This assignation is based on the presence of an extended hyperbolar zone on the gladius, which also lacks the keel that is otherwise typical of contemporary teudopseid octobrachians. However, a more precise determination is hampered by the fragmentary preservation, and the diagnostic course of the growth lines within the hyperbolar zone is not observable on this specimen. Fossils of loligosepiids are widespread in the Lower Toarcian sediments across Central and Western Europe and have been described for instance from Southern Germany, France, Luxembourg and the UK (*e.g.*, *Riegraf, Werner & Lörcher, 1984*; *Doyle, 1990*; *Guérin-Franiatte & Gouspy, 1993*; *Fuchs & Weis, 2008*; *Fuchs & Weis, 2010*; *Jattiot et al., 2024*). In the Schistes bitumineux Formation, they usually have phosphatised soft tissue and ink sacs, in addition to the unmineralised gladius. The vampyromorph faunal list in

this formation comprises three teudopsid species (*Teudopsis bunelii, Teudopsis subcostata* and *Teudopsis bollensis*) and three loligosepiid species (*Loligosepia aalensis, Parabelopeltis flexuosa, Jeletzkyteuthis coriaceus*) as well as one recently discovered species, *Simoniteuthis michaelyi* (*Fuchs & Weis, 2008*; *Fuchs & Weis, 2010*; *Fuchs, Weis & Thuy, 2024*).

### Functional anatomy of coeval octobranchian-eating pachycormids

Several species of pachycormids are found in the same deposits as MNHNL TV221: *Pachycormus macropterus, Saurostomus esocinus, Sauropsis latus, Euthynotus incognitus*, and *Haasichthys michelsi* (*Delsate, 1999a*; *Delsate, 1999b*) but only the former two have coleoid remains in their gut content (*Weis et al., 2024*). Both are smaller than *Stenopterygius triscissus*, with skull lengths of around 150 mm for *Saurostomus* (of which mandible length accounts for 45–50%; total body length up to 1,374 mm (*Friedman, 2012*; *Cawley et al., 2019*) and 170 mm for *Pachycormus* (of which mandible length accounts for 40–55%; total body length up to 600 mm (*Friedman, 2012*; *Cawley et al., 2019*; *Cooper & Maxwell, 2022*)). *Pachycormus macropterus* and *Saurostomus esocinus* have a highly hydrodynamic, fusiform profile, suggesting rapid swimming. This is also supported by the aspect ratios of the pectoral and caudal fins, which are fairly similar for *Stenopterygius* and the three pachycormids considered (Fig. 3).

Besides these similarities in global body shape and fin aspect ratios, the ichthyosaurian and the pachycormids fundamentally differ in the shape and mechanism of their feeding apparatus. Firstly, *Pachycormus* and *Saurostomus* lack a rostrum; the anteorbital portion of their skull is short. This results in marked differences in the closing mechanical advantage, which is $\approx 0.22$ for *Saurostomus esocinus,* and 0.25 for *Pachycormus macropterus*, therefore nearly or more than three times that of *Stenopterygius triscissus* (Fig. 3). Similarly, the mandible aspect ratio is about five times higher in *Saurostomus esocinus* and *Pachycormus macropterus* than in *Stenopterygius triscissus* (Fig. 3). This suggests that *Pachycormus macropterus* and *Saurostomus esocinus* did not employ lateral snapping, but rather suction to aid capturing prey items. The tooth crowns of *Pachycormus macropterus* are small and elongated (crown shape ratio = 2.7; *Friedman, 2012*), with a slight labio-lingual compression (*Cooper & Maxwell, 2022*), and lack any evident external texture (although 3 $\mu$m-wide apicobasal ridges are present; D. Delsate, pers. obs., 2024). Tooth crowns of *Saurostomus esocinus* are larger, and less elongated (crown shape ratio = 1.84 (*Friedman, 2012*), and curved lingually (*Cooper & Maxwell, 2022*), with conspicuous folds in the enamel. *Pachycormus macropterus* and *Saurostomus esocinus* possess pupils that are absolutely smaller than those of *Stenopterygius triscissus* (18 mm and 21 mm *vs.* 36 mm; Fig. 3), resulting in a vision range of a 10 cm black target at a depth of 350 m in clear oceanic waters ($\approx 23$ m (*Nilsson, Warrant & Johnsen, 2014*)), while it was $\approx 32$ m for *Stenopterygius triscissus*).

## DISCUSSION

### Gut content in Early Jurassic ichthyosaurians

Gut contents have been reported in ichthyosaurians, mostly in the Early Jurassic medium-sized thunnosaurians *Ichthyosaurus* and *Stenopterygius* (*Pollard, 1968*; *Massare, 1987*;

*Dick, Schweigert & Maxwell, 2016*). *Buckland* (*1829*, p. 226) inferred that Early Jurassic ichthyosaurians preyed upon cuttlefish (suborder Sepiida; crown Decabrachia), as he regarded ring-like structures in putative ichthyosaurian coproliths as the horny sucker rings of sepiids. The occurrence of sepiids in the Jurassic has however been repeatedly rejected ((*Fuchs, 2023*) and references therein), because fossils with a minimum set of sepiid characters are recorded from the latest Cretaceous onward, thus well after the extinction of ichthyosaurians (*Bardet, 1992*; *Fischer et al., 2016*). Subsequently, the idea grew that hooklets associated with ichthyosaurian remains belonged to squids (suborder Oegopsida; crown Decabrachia) such as the present-day *Onychoteuthis* (*Moore, 1856*). The idea of 'fossil teuthids' was at that time popular, but has become obsolete since the works of *Bandel & Leich (1986)*; *Fuchs (2020)*. The fossilised diet of *Ichthyosaurus* appears to be restricted to belemnoid hooklets (*Pollard, 1968*; *Massare, 1987*) and without much variation (nor more details) among species. Two specimens are also known to preserve teleost scales (*Buckland, 1836*; *Lomax, 2010*), along with multiple belemnoid hooklets. *Keller (1976)* and *Massare (1987)* reported belemnoid hooklets and rare fish fragments in *Stenopterygius quadriscissus*, *Stenopterygius crassicostatus* (=*Stenopterygius quadriscissus* or *Stenopterygius uniter*), and *Stenopterygius megalorhinus* (=*Stenopterygius triscissus* or *Stenopterygius uniter*) (*Maisch, 2008*). The diet of the species *Stenopterygius quadriscissus* seems to follow an ontogenetic trend increasing the proportion of cephalopods while decreasing that of teleosts (*Dick, Schweigert & Maxwell, 2016*). Individuals with mandible length >420 mm seemingly solely relied on cephalopods; which corresponds to the moment when teeth reduce in relative size, eventually becoming non-functional (*Dick & Maxwell, 2015*). In *Stenopterygius triscissus*, teeth remain large enough to protrude from the dental groove at larger skull sizes (*Maisch, 2008*; this work). Larger species of Early Jurassic ichthyosaurians seem to add amniotes to their diet. Indeed, fragments of a smaller ichthyosaurian have been reported in the gastric content of '*Leptopterygius acutirostris*' (*Massare, 1987*) (some specimens of that entity are now referrable to the large parvipelvian *Temnodontosaurus zetlandicus*, while some smaller specimens possibly belong to early baracromians *Maisch, 2010*; *Laboury et al., 2022*). *Böttcher (1989)* reported 200 ichthyosaur centra (as well as a series of coleoid hooklets) in the gastric mass of '*Leptopterygius burgundiae*' (=*Temnodontosaurus trigonodon*) (*Maisch, 1998*; *McGowan & Motani, 2003*; *Bennion et al., 2024*). *Serafini et al. (2025)* markedly expanded the knowledge of ichthyosaurian predation by *Temnodontosaurus trigonodon*, reporting several cases of ingestion of specimens of *Stenopterygius*, in addition to belemnitid hooklets.

However, there has been confusion, or perhaps a lack of precision, in the literature regarding the possible owners of hooklets found in ichthyosaurian guts. These claw-like structures were interpreted as the arm hooklets of 'belemnites' (*Pollard, 1968*) and this somehow became the interpretation by default. In fact, the coleoid order Belemnitida is essentially typified by ten hooklet-bearing arms (*Fuchs & Hoffmann, 2017*), and a massive calcitic rostrum covering the primary shell, as well as the presence of mega-onychites (*Hoffmann & Stevens, 2020*). The absence of such rostra in the stomach of ichthyosaurians was longtime enigmatic until *Valente, Edwards & Pollard (2010)* suggested that most of the arm hooklets in the stomachs of Early Jurassic thunnosaurians do not belong

to 'belemnites' *sensu stricto*, but rather to rostrum-less clades, essentially referrable to Phragmoteuthida, Belemnoteuthina, and Diplobelida (*Fuchs, Donovan & Keupp, 2013*; *Fuchs, 2019*). So far, unambiguous evidence of ichthyosaurians feeding on belemnitids is rare; sporadic records of mega-onychites (*Dick, Schweigert & Maxwell, 2016*) suggest that Early Jurassic thunnosaurians occasionally preyed upon rostrum-bearing belemnites, potentially snapping the hard part off (*Valente, Edwards & Pollard, 2010*), but preferentially hunted rostrum-less coleoids. However, larger taxa such as the non-thunnosaurian parvipelvian *Temnodontosaurus* consumed belemnitids whole (*Serafini et al., 2025*).

## Competition or chance?

The fossil record of octobrachian coleoids as ichthyosaurian gut content is exceedingly rare; the specimen we described above represents the first evidence of this behaviour, even though *Dick, Schweigert & Maxwell (2016)* noted that these species may have been part of ichthyosaurian diets. The fossil record of octobrachians as gut content has been regarded as certainly biased, because octobrachians usually lack strongly mineralised tissues and hooklets (*Dick, Schweigert & Maxwell, 2016*). The fact that none are preserved in German specimens of *Stenopterygius quadriscissus* prompted *Dick, Schweigert & Maxwell (2016)* to suggest that this species avoided octobrachians. With hundreds of specimens known (*Hauff, 1953*), *Stenopterygius quadriscissus* dominates the marine reptile fossil record in early Toarcian German localities and is usually regarded as the ideal representative of the ecology of *Stenopterygius*. Yet, *Stenopterygius triscissus* is more common in Luxembourg (*Godefroit, 1994*; *Godefroit, 1996*) and only 15 German specimens of *Stenopterygius triscissus* have been reported with gut content (*Dick, Schweigert & Maxwell, 2016*). Moreover, there are subtle differences in the gross anatomy (*Maisch, 2008*; *Maxwell, 2012*) and in the retention of functional teeth at adult size (*Dick & Maxwell, 2015*; *Dick, Schweigert & Maxwell, 2016*) between the two species, which could lead to slightly distinct diets.

Fossilised gut contents represent a biased record of the diet of marine predators, whose fossil record is also strongly biased (*e.g.*, *Benson et al., 2010*). It is therefore difficult to assess the biological significance of our find, as a frequent or infrequent ingestion of easily dissolved prey would result in fossil records that are difficult to dissociate. Therefore, it is presently impossible to infer whether the specimen we describe here documents a clear case of interclass competition between *Stenopterygius triscissus* and coeval, loligosepiid-eating pachycormid teleosts or just a rare behaviour. Nevertheless, it is now certain that these species ate loligosepiids, which prompted us to visualise how mechanically distinct these predators are (Fig. 3). These size and shape differences are further reinforced by the complexity of the jaw opening and closing in teleosts, as the maxilla and the ceratohyals can rotate, markedly widening the bucco-pharyngeal cavity. This adds suction as an essential mechanism in prey capture (*e.g.*, *Day et al., 2015*). Closing the mouth forces water to exit behind the opercula, causing a stronger water flux (pushing the prey) at the entry of the oesophagus than in reptiles. This enabled pachycormids to capture and consume relatively larger prey items than ichthyosaurians did. Indeed, *Pachycormus* is known to feed on juveniles of its own kin (*Cooper, 2023*) as well as ammonites (albeit possibly accidentally) (*Cooper & Maxwell, 2023*). Such profound functional differences between

marine amniotes and 'fish' result in the absence of a 'convergence requirement' to feed on similar prey items in the marine realm, which likely explains why attempts to unify the feeding guilds of marine amniotes and sharks have been challenging (*Ciampaglio, Wray & Corliss, 2005*), despite palaeontological and neontological evidence that both groups competed for similar resources (*e.g.*, *Martin et al., 2017*). Exceptional preservation in the early Toarcian Bascharage Lagerstätte of Southern Luxembourg opens up this possibility for *Stenopterygius triscissus*, *Pachycormus macropterus*, and *Saurostomus esocinus*.

## CONCLUSIONS

We report on the gut content of a specimen of the ichthyosaurian *Stenopterygius triscissus* from the early Toarcian Bascharage Lagerstätte of Southern Luxembourg. The gut contains an identifiable gladius of a loligosepiid vampyromorph; this marks the first record of gladius-bearing octobrachians in the diet of ichthyosaurians. The coeval pachycormid teleosts *Pachycormus macropterus* and *Saurostomus esocinus* have also been reported to feed on the same food resource, although through a distinct mechanism, given the functional differences between these taxa. The intensity of the possible competition between ichthyosaurians and pachycormids during the Early Jurassic remains, however, elusive.

**Institutional abbreviations**

| | |
|---|---|
| **BRLSI** | Bath Royal Literary and Scientific Institution, Bath, UK |
| **GPIT** | Paläontologische Sammlung der Universität Tübingen, Germany |
| **MNHNL** | Palaeontological collections of Musée National d'Histoire Naturelle, Luxembourg |
| **NHMUK** | Natural History Museum, London, UK |
| **SMF** | Natur-Museum Senckenberg, Frankfurt, Germany |
| **SMNS** | Staatliches Museum für Naturkunde Stuttgart, Germany |

## ACKNOWLEDGEMENTS

We are indebted to Paul Braun (Musée National d'histoire Naturelle, Luxembourg), who took the photographs of the specimen and prepared the 3D model. We warmly thank editor Dagmara Żyła and reviewers Elizabeth Haper, Feiko Miedema, and one anonymous colleague for their constructive and thoughtful comments; they played a clear role in making this paper better.

### Funding

The authors received no funding for this work.

### Competing Interests

The authors declare there are no competing interests.

## Author Contributions

- Valentin Fischer conceived and designed the experiments, analyzed the data, prepared figures and/or tables, authored or reviewed drafts of the article, and approved the final draft.
- Robert Weis conceived and designed the experiments, analyzed the data, prepared figures and/or tables, authored or reviewed drafts of the article, and approved the final draft.
- Dominique François Delsate performed the experiments, authored or reviewed drafts of the article, and approved the final draft.
- Francesco Della Giustina performed the experiments, analyzed the data, authored or reviewed drafts of the article, and approved the final draft.
- Pierre Wintgens performed the experiments, analyzed the data, prepared figures and/or tables, and approved the final draft.
- Dirk Fuchs analyzed the data, authored or reviewed drafts of the article, and approved the final draft.
- Ben Thuy conceived and designed the experiments, prepared figures and/or tables, authored or reviewed drafts of the article, and approved the final draft.

## Data Availability

The data (measurements, ratios) and code (R scripts) are available in the Supplementary Files.

## Supplemental Information

Supplemental information for this article can be found online at http://dx.doi.org/10.7717/peerj.19786#supplemental-information.

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
