# Peer review of "Vampyromorph coleoid predation by an ichthyosaurian from the Early Jurassic Lagerstätte of Bascharage, Luxembourg"

_PeerJ, doi:10.7717/peerj.19786_

## Round 0.1 · original submission · Major Revisions

Please, take into account all reviewers' comments and suggestions.

Reviewer 1 ·

Basic reporting

no comment

Experimental design

no comment

Validity of the findings

no comment

Additional comments

Gut content of Early Jurassic ichthyosaurs is important to understand the feeding ecology and structure of the marine ecosystem. It is well known that Jurassic ichthyosaur Stenopterygius were fast swimmer, their body shape and anatomy were adapted to hunt soft and fast prey items such as cephalopods, but the detailed information about the prey items from the gut content was poorly known. This paper describes the ecomorphology and the gut content of a specimen of Stenopterygius triscissus from the Bascharage Lagerstätte, compares it with the coeval predators such as pachycormid teleost Pachycormus macropterus and Saurostomus esocinus, feeding on the same food resource (loligosepiid vampyromorph coleoid) but through a fully distinct mechanism. It helps to better know the ecology of the Early Jurassic ichthyosaurs.

There are two points that need to be addressed:
1. P9, line 91-93: “The specimen had been described and illustrated as Stenopterygius quadriscissus by Godefroit (1994: pl. 3, fig. 93c)”, but on P11, line 151-152: “This suite of features suggests that the specimen belongs to the species Stenopterygius triscissus (Godefroit, 1994; Maisch, 152 2008; Maxwell, 2012). It looks confusing, please clarify this point.
2. On pages 4 and 5, the sequences of the authors appear different; please also clarify this.

·

Basic reporting

Well written, well referenced, adequate data and images

Experimental design

OK
Though overstretched - see report

Validity of the findings

Overstretched - see report

Additional comments

This is an interesting specimen, and this manuscript will make a valuable contribution to the literature on food webs in the Jurassic. It seems very clear that coleoid gladius has been ingested one way or another by the ichthyosaur. The identification of the predator and ‘prey’ is well established. As such, this is the first report of an ichthyosaur eating an octobrach. And that is important, and it makes the paper very citable.

But I think the contribution goes too far.

The text itself notes the rarity of this finding, and of course, there are taphonomic explanations that suggest why this may be so, even if these squids were a major part of the diet. Line 329 ff states, “However, if octobrachian consumption was not uncommon in Stenopterygius triscissus, it would document a case of competition among dissimilar predators.” That IF is a bit crucial! Alas, we don’t know that this was an intentional predation, nor if such a selection was common. And that is a huge pity, but there is no scientific support for making a bigger thing out of it.

Just because there is also evidence that pachycormid fish also, on occasion, ate coleoids is no reason to suggest that they competed. To do so would suggest that 1) the food source was in short supply and 2) they genuinely occupied the same habitat. There is no actual
evidence for either.

The data in Table 1 shows a baffling degree of precision! These ratios do not need to be given to the nearest 0.000000001!

I have no wish or need to remain anonymous.

Liz Harper

·

Basic reporting

Basic reporting:
Language: The language is very good and appropriate for the reader. No need for extra clarifications or English language checks. I annotated a few things in the pdf file, but nothing major
Structure: Seems fine to me. The paper is clearly set-up in the usual manor for a paleontological research paper.
Intro + references: The introduction has an appropriate length. The literature cited is relevant to the study. I did ask for a few clarifications and extra references at certain points over the manuscript, but nothing major.
Figure relevance/clarity: All figures are relevant in my opinion, although their integration and clarification could be improved on to a certain extent. Figure 2 for example is never referred to in the text. Figure 1 has a very brief description (essentially only the figure title). Even though I appreciate that many people know where Luxembourg is located, it might still be nice to spend a few words describing the main details of the maps you provide. Likewise in figure 3, you can’t just refer to table 1 for details. In my opinion a brief description of all the parameters used needs to be present in the figure description.

Experimental design

Experimental design:
Originality: This is original research and the manuscript provides new data. In my opinion the results fit within the scope of the journal
Research question well-defined: The research question is well defined, but I am uncertain on whether the measurement data provided actually change the discussion and conclusion of the paper. Do not take this the wrong way, as I appreciate all the work that has gone in to the data collection, but isn’t it sort of obvious that even though Stenopterygius and several pachycormids may feed on similar prey, their feeding techniques must differ? The differences in cranial and mandibular proportions and the hyoidal differences were (at least to me) also obvious without the extra data. Maybe, it is good that someone actually does the math and shows the discrepancy and my point might therefore be mute. I leave this up to the editor’s discretion.
Conduction of research and methods: Methods seem to be well-described and well-conducted to me. No criticism here.

Validity of the findings

Validity of findings:
The findings are relevant to the ecology and behaviour of several extinct species and relevant to the European Toarcian marine ecosystem as a whole. They are definitely beneficial for future researchers interested in predator-prey interactions and ecological tiering and niche partitioning in the Early Jurassic ocean. In my opinion this paper is therefore likely to be cited in wider studies on Toarcian marine ecology and is therefore of benefit to the literature. The data provided are robust, held at relevant institutions open to researchers are therefore replicable and transparent.
Conclusions well grounded
I like that the authors are very nuanced in their discussion section on dietary preference in any taxa discussed. Therefore, their conclusions seem minor, but are grounded very much in reality!

Additional comments

Stenopterygius specimen: I think the determination of species can be more precise. Maxwell (2012) did a metrics study that differentiated the different taxa quite well. Would be interesting to see where this specimen plots, I think most of the relevant data are there, so it is worth a try (and is quite easy to do). I want to stress that the determination of S. triscissus is probably correct at a glance, but it might be good to have this backed up by data. Secondly, the authors go into a discussion on ontogenetic stage, but unfortunately, they do not discuss the relevant work done recently on the very closely related species S. quadriscissus. I applaud the authors for including an ontogenetic section, as it is relevant and many descriptive manuscripts lack such a section. However, given the recent literature, this section could be improved on.

---

## Round 0.2 · accepted · Accept

Thank you for addressing all reviewer's comments. I am happy with the current version and recommend it for publication.

·

Basic reporting

Overall comments:
I am very happy with the new manuscript. The authors have taken the reviewers comments to heart and improved their manuscript majorly. Especially the re-assessment of the species determination will spark further research into separating species of Stenopterygius. I recommend no further revisions and hope to see this published soon.


Basic reporting:
Language: No further comments
Figure relevance/clarity: I am happy with the changes the authors made in order to integrate the figures into the manuscript. The figures captions are likewise now appropriate. Many thanks!

Experimental design

Experimental design:
Originality: This is original research and the manuscript provides new data. In my opinion the results fit within the scope of the journal
Supplemental files: I have assessed all supplemental data and observe no discrepancies or oddities.

Validity of the findings

Validity of findings:
I am happy with the further nuance and extra acknowledgements the authors have given regarding the comparison between the coeval teleosts and marine reptiles. I am also still happy the authors decided to include the numerical data and their explanation. It is good that classical paleontology becomes a degree more quantitative.

Additional comments

Stenopterygius specimen determination and ontogenetic stage:
The authors did a great job integrating my previous comments in this regard. They even managed to slightly improve and correctly criticize previously proposed methodology of separating the species of Stenopterygius. I regard the manuscript to be much more valuable due to this. I hope this sparks further improvement on separating Stenopterygius species (if possible) as many new specimens from different localities are now coming to the forefront. I am also very happy with their improved assessment of ontogeny; it now includes all relevant previously published information. The re-assessment even changed their ontogenetic identification slightly.